# Human–Animal Interactions with *Bos taurus* Cattle and Their Impacts on On-Farm Safety: A Systematic Review

**DOI:** 10.3390/ani12060776

**Published:** 2022-03-19

**Authors:** Frances Margaret Titterington, Rachel Knox, Stephanie Buijs, Denise Elizabeth Lowe, Steven James Morrison, Francis Owen Lively, Masoud Shirali

**Affiliations:** 1Agri-Food and Biosciences Institute, Large Park, Hillsborough BT26 6DR, UK; frances.titterington@afbini.gov.uk (F.M.T.); stephanie.buijs@afbini.gov.uk (S.B.); denise.lowe@afbini.gov.uk (D.E.L.); steven.morrison@afbini.gov.uk (S.J.M.); francis.lively@afbini.gov.uk (F.O.L.); 2AgriSearch, Innovation Centre, Large Park, Hillsborough BT26 6DR, UK; rachel@agrisearch.org

**Keywords:** injury, behaviour, management, handling, facilities

## Abstract

**Simple Summary:**

Cattle are large animals that can cause serious injuries to humans. Humans may encounter cattle through working on farms, living on a farm, or traversing fields with cattle. A systematic review was carried out to assess the factors which may lead to a dangerous interaction with cattle. A literature search was carried out to find papers that included the criteria ‘Bovine’, ‘Handling’, ‘Behaviour’ and ‘Safety’, or terms therein. The search returned 17 papers, and after collation, six themes were identified: actions of humans; human demographics, attitude, and experience; facilities and the environment; the animal involved; under-reporting and poor records; and mitigation of dangerous interactions. Exploration of these themes shows that more accurate recording of interactions before an injury is required. Furthermore, targeted, tailored education for anyone who may come into contact with cattle could reduce cattle-induced injuries.

**Abstract:**

Cattle production necessitates potentially dangerous human–animal interactions. Cattle are physically strong, large animals that can inflict injuries on humans accidentally or through aggressive behaviour. This study provides a systematic review of literature relating to farm management practices (including humans involved, facilities, and the individual animal) associated with cattle temperament and human’s on-farm safety. The Preferred Reporting Items for Systematic reviews and Meta-Analyses (PRISMA) was used to frame the review. Population, Exposure, and Outcomes (PEO) components of the research question are defined as “Bovine” (population), “Handling” (exposure), and outcomes of “Behaviour”, and “Safety”. The review included 17 papers and identified six main themes: actions of humans; human demographics, attitude, and experience; facilities and the environment; the animal involved; under-reporting and poor records; and mitigation of dangerous interactions. Cattle-related incidents were found to be underreported, with contradictory advice to prevent injury. The introduction of standardised reporting and recording of incidents to clearly identify the behaviours and facilities which increase injuries could inform policy to reduce injuries. Global differences in management systems and animal types mean that it would be impractical to impose global methods of best practice to reduce the chance of injury. Thus, any recommendations should be regionally specific, easily accessible, and practicable.

## 1. Introduction

Working with cattle is widely reported as a major cause of human injury on the farm [1,2,3]. This is a global problem, with cattle cited as the cause of injury in 127 hospitalisations in the midland region of New Zealand over a five-year period [4] and 221 cattle-induced injuries reported over a seven-year period in the United States [5]. In England, one hospital reported sixty-seven patients that were admitted over a five-year period following cattle-related accidents [6]. However, it is difficult to quantify actual numbers of cattle-induced injuries, and hospital admissions under-report as cattle-induced injuries are often treated by a General Practitioner [4]. Cattle-related injuries can be severe, with 24 people reported to have died from injuries sustained from cattle between 2015 and 2020 in Great Britain [7]. In the Republic of Ireland, cattle were in the top five triggers for fatal accidents at work, accounting for seven fatal accidents [8]. Collating data on injurious interactions with cattle is difficult because official statistics often group injuries or fatalities, and as a result data is low resolution and can be difficult to interpret. Official statistics report numbers of fatalities or reported injuries, but do not specify the behaviour of the animal or human before the injurious event [7,8]. In order to prevent injuries and potential fatalities due to livestock, it is important to identify contributing factors that may result in cattle reacting in a dangerous manner.

Livestock handling is a complex interaction between people, animals, and the environment [9]. Research into the person involved, the handler, is limited and disparate [10]. The person involved can induce fear reactions in animals, in turn reducing productivity and welfare [11,12]. It is reported that animals that are handled negatively may react adversely and sustain injuries [11]. Negative consequences of fear induced by bad handling may reinforce the handlers’ negative attitudes and actions towards animals [13]. The human’s attitude affects the interaction and how the animal is handled [14], with handlers who had a better attitude behaving in a manner which positively influenced animal behaviour [15]. In addition to the human element, the animals’ behaviour, personality, or temperament will affect human–animal interactions. The animals’ general behaviour and reaction to humans can be defined as their temperament [16]. Temperament can be affected by environmental factors, such as previous handling experiences [17] and animal genetics [18,19]. Although temperament is a subjective trait that is difficult to measure, farmers are reported to be aware of differences in temperament between individual animals [20]. The third aspect is the environment, with some cattle handling injuries directly related to farm infrastructure [21]. Improving farm infrastructure can improve profitability and safety [22] and reduce stress in animals [23]. However, an American survey found that farmers can be reluctant to upgrade handling facilities due to the large investments of time and money required [21].

Although the relationships between animal genetics with animal behaviour and farm management with animal behaviour are well researched, there are few studies on the specific human actions which may reduce human safety. This paper aims to identify the common factors in incidents where cattle have had an injurious or potentially injurious interaction with a human. Knowledge of the factors leading to injury can be used to direct policy and aid decision makers on farm to develop protocols to reduce the number of injuries to humans involving livestock. The objective of this paper is to conduct a systematic review of the literature with the aim of identifying specific human–cattle interactions and management processes that may increase the likelihood of unsafe interactions with humans on farms. This review is designed to address on-farm safety and includes all potential animal encounters on-farm (including visitors and on-farm workers); it will not include animals bred for sport or fighting. To ensure this review can fulfil the aim of informing policy, papers will be limited to those published since the year 2000. The occurrence of fatalities involving animals have shown a marked increase in the legislative area (Northern Ireland, United Kingdom) in recent years [24] so only technologies relevant to modern farm managers and modern animal handling environments will be included. Additionally, the cattle will be limited to *Bos taurus* species. *Bos indicus* and their crosses are reported to be more reactive than *Bos taurus* cattle [25]; this review focuses on *Bos taurus* as this is the most populous breed in the target legislative area. The outcomes of this review will be used to develop guidelines and identify where improvements in animal handling need to be made.

## 2. Materials and Methods

### 2.1. Eligibility Criteria

In order to undertake a systematic review, it is necessary to identify the Population, Exposure, and Outcomes (PEO) components of the research question [26]. Expert opinion from authors was considered, and a list of keywords relevant to the topic was developed. Preliminary searches of the Web of Science database using the keywords were carried out to assess the number and quality of papers returned. Search terms were discussed and agreed upon by all the authors and subsequently grouped into search categories. The population was defined as “Bovine”, with an exposure of “Handling” and outcomes of “Behaviour” and “Safety”. The search could not be pre-registered as the outcomes did not directly impact human health and so the search was not within scope for a designated public repository.

### 2.2. Information Sources

The procedure for undertaking this review was designed following the PRISMA framework [27]. Three databases were identified as suitable for searching: PubAg, Web of Science, and the ‘EBSCO Academic search elite’ option within the research database of EBSCO. Searches were limited to papers published between the year 2000 and the date of search (June 2021).

### 2.3. Search Strategy

The following algorithm of keywords was designed, where results must have at least one search term from each PEO component: (cattle or cow or “steer” or heifer or bull or dairy or beef or herd or “*Bos taurus*”) AND (hand* or attitude or manage* or farmer or human or automat*) AND (temperament or excit* or aggressi* or fear* or docility or flight or “exit score” or “exit time” or “exit speed” or “exit velocity” or “chute score” or “strain gauge” or “movement measuring device” or personality or “coping style” or boldness or proactive or reactive) AND (safe* or injur*). An asterisk denotes a truncated term with ‘wildcard’ which may represent one or multiple characters, and inverted commas instruct the search engine to find an exact match to the term. Searches were conducted by a single author and validated through an independent replication of the search by another author.

### 2.4. Selection Process

In total, 694 references were returned and exported as .csv files. Fifty-two duplicate texts were removed, then the process of title and abstract screening was carried out independently by two authors. Two articles were conference abstracts that could not be obtained through the institution’s current subscriptions, through interlibrary loans, or after contacting the original author and thus were discarded. General narrative reviews were discarded but used as sources of potentially relevant studies. Eligibility criteria was set whereby the studies must fulfil the following: (1) have *Bos taurus* as subject; (2) include farm management/human interaction; (3) have an animal response that can potentially negatively impact handler safety, i.e., a reaction which is identified as dangerous, threatening or risky; (4) be written in English (5); have the full text available. Papers were limited to those published in English as this was the working language of the authors. Once the screening sift was complete, any differences in opinion between the two reviewers were discussed and resolved. The number of full texts for eligibility assessment was 48. Of these, 17 relevant papers were included in the review. A full breakdown of the sifting process is shown in Figure 1.

### 2.5. Synthesis of Results

Once suitable papers were identified, they were tabulated to compare study type, design, animals used, and the outcomes (Appendix A). The journal ranking of each publication was not considered an appropriate measure of credibility as this reflects the quality of the journal rather than the articles therein [28]. Due to the publishing dates of the papers included (ranging from 2009 to 2021), it would not be possible to accurately compare citation count. As a result, papers were evaluated by two authors independently using the Critical Appraisal Skills Programme (CASP) checklists [29]. It was not possible to collate results for meta-analysis as the data were heterogeneous and information from different papers could not be directly compared. Data were summarised by author, year, geographical location, animal breed type, sample size, measurement of exposure, study aim, findings and recommendations, and any limitations of the study. Once data was collated, a thematic analysis was carried out which identified similar themes and topics within the study aims, measurements of exposure, and findings.

## 3. Results and Discussion

### 3.1. Study Selection

The principal reason for discarding papers was that the subject of the study was not *Bos taurus*, this included studies pertaining to *Bos indicus* or other species of animal which used the noun “Cow” for females. A large number of food supply chain studies were returned in the search due to the inclusion of the terms “dairy”, “beef”, and “safety” in the protocol. Papers which recorded either injury or handling but did not record any interaction between the two. Two studies were removed because they were not studies of farmed cattle, the subjects of these studies were cattle bred for rodeo or fighting.

### 3.2. Study Characteristics

The 17 papers identified as meeting the criteria were appraised and considered suitable for inclusion. The papers predominantly assessed dairy breeds (*n* = 8), with four papers assessing beef cattle and five papers were undefined breeds within the *Bos taurus* species. Three non-narrative review studies were included which assessed media reports [30], online records, and on-farm injury reports [31] or hospital records [32]. These reviews were considered suitable for inclusion as they were non-narrative reviews that collated public records and followed a search protocol. Five papers were qualitative assessments of opinions gathered through interviews or focus groups. Participants included farmers who had participated in safety campaigns [33], used organic dairy systems [34], had a specific breed of cattle [35], veterinarians who carried out castration procedures [36], and immigrant workers who worked with dairy cattle [37]. Four studies were observational studies that monitored maternal temperament of beef cattle [38], habituation of primiparous cows to milking procedures [39], environmental effects such as restraint [40], and outdoor exercise [41] and their effects on the human–animal relationship. The remaining five studies combined observations of animal behaviour and questionnaires or surveys of farmer opinions [42,43,44,45,46]. Two of the articles were by the same author and on examination of the data were found to be two different analyses based on the same study, which included a small number of dairy farms [44,46]. The studies had a wide geographical spread, but were predominantly European (n = 10), with four North American studies and one Indian study. Two studies included reviews of international data. All were published between 2009 and 2021.

### 3.3. Summary of Evidence

The 17 papers identified a range of management practices that could affect, either positively or negatively, the occurrence of risky animal interactions and explored how both human actions and attitude can affect the occurrence of cattle attacking. An important aspect was the human involved and their behaviour through acting calmly, having a positive attitude to both cattle and risk, and assigning adequate time and facilities to undertake tasks on the farm. An animal’s propensity to attack could be affected by the animals’ previous experiences, inherent traits for disposition, and previous experiences, with human–animal interactions that were positive and not stressful for the animal. These were organised into six themes: actions of humans; human attitude and experience; facilities and the environment; the individual animal; underreporting; and current forms of mitigation.

### 3.4. Themes

#### 3.4.1. Actions of Humans

Farmers feel that they can positively impact animal behaviour through their own actions. Farmers reported that staying calm, using personal experience, and talking to the animals could facilitate cattle handling [43], and provisioning enough time for a task can reduce the likelihood of injury [37]. Positive experiences for the animal during human interactions, for example, the human providing concentrates directly to the animal rather than using a machine, resulted in significantly calmer cows measured by Qualitative Behaviour Assessment [42]. When working with animals, it is important to consider how they will react in a procedure that may be averse. A survey found that 37% of US veterinarians considered the risk of injury to the operator critically important when performing castration, a painful procedure [36]. An increase in cows slipping, falling, or cow behaviours indicating fear (collectively referred to as risk situations) occurred more when dairy cows were moved to hoof trimming than to milking [44,46], which may be because hoof trimming is a more aversive procedure. However, it can be difficult for farmers to judge how aversive an animal may find a procedure. When farmers were asked to rate their agreement (on a scale of 1 to 5; where 1 is “totally disagree” and 5 is “totally agree”) to the statement “animals experience physical pain as humans do” there was a wide variation in responses [45].

Despite the causal relationship not being clear, quiet handling and minimal talking are recommended to reduce risk situations with cattle [46]. Forceful tactile interactions using an object should be avoided as they were found to elicit fear reactions from cattle and could be counterproductive, slowing down procedures by causing the animal to freeze [46]. Human interactions which led to potential injury included pulling the halter, which was positively correlated with being head-butted, and use of forceful interactions such as shouting, hitting, or tail twisting for an extended duration, which increased the likelihood of the handler getting kicked [46]. Time spent in the risk zone, i.e., within reaching distance of the cow, was correlated to the number of observed risk situations [46]. This increase was attributed to close proximity to the animal, leading to an increased risk of being injured by an unexpected response or reaction by the animal [44]. Conversely, farmers who made physical contact with cattle during monitoring tended to have herds with lower avoidance distances [43], suggesting that contact and close proximity to an animal can improve interactions.

There was a paucity of literature on specific human actions which directly caused injurious interactions between cattle and humans. This is because the injury was reported by the injured party and generally did not clearly define the precise interactions which led to an injury [31,32] and may have created bias. Resultantly, many of the papers considered animal reactions that were potentially injurious rather than those which caused an injury to humans. Without records of every occurrence of each potentially dangerous behaviour, it is not possible to estimate how often these reactions lead to an injury, thus it is important that the events leading to animal inflicted injury are accurately reported and recorded. Collating this data can allow thorough analyses of specific actions leading to cattle-inflicted injuries.

#### 3.4.2. Human Demographics, Attitude, and Experience

Human actions will be dependent on a range of factors, including their demographics attitude, and experience. A questionnaire amongst Swedish handlers that identified the handler’s risk tolerance found that risk-accepting handlers encountered significantly fewer risk situations per minute when moving cows to hoof trimming than risk-averse handlers [44]. However, no correlation between the handler’s attitude towards cows and risk situations was found [44]. In contrast, cows in herds with managers that agreed more strongly with the importance of positive animal contacts were significantly calmer than cows in herds with managers who did not believe this [42]. More evidence of the human attitude adversely affecting cattle behaviour was expected. In the papers reviewed, farmer attitudes were often assessed through a survey or focus group—these are qualitative studies that are difficult to validate [47]. Some respondent groups were limited to certain breeds [35] or management [34] and the results may be less transferable to different management systems. In all cases, the focus groups/interviewees agreed to take part in the study, meaning respondents may have been more progressive than those who refused, or otherwise not representative of the general population.

No study could identify a strong association between the handler demographic and handler safety. Although the majority of reported bull attack victims were middle-aged males, it was noted that this was consistent with the predominant demographic of US farmers [31]. It was reported that handler demographics did not affect interactions with cows, however, this study had a small cohort (12 handlers), with a high proportion (75%) of males [44]. Although no associations could be proven, it was suggested that youths should not work with bulls as they lack the necessary maturity and strength [31]. In a survey of Swedish farmers, it was found that older farmers thought they could mitigate risks by being more careful, however, it was found that their younger counterparts did not think that older farmers were careful enough [33]. This is a theme throughout the survey, with farmers fearing risks to third parties such as children, the elderly, and workers on the farm rather than themselves [33]. The issue of communication between workers, particularly migrant workers, was highlighted. Managers may have communication difficulties with migrant workers, meaning that they have limited control over staff [33], however, a survey of migrant workers blamed managers for not educating migrant workers about risks on the farm [37]. This highlights that farm safety is an issue that is not only pertinent to the individual, but the farm manager and risk assessments and education programmes must also reflect this.

Most injuries recorded are from people who worked with cattle, such as farmers and vets [30]. This aligns with the finding that injury risk is associated with hours of exposure to an animal [31]. Conversely, people who are visiting a farm and unexpectedly come in contact with cattle may have limited knowledge of how to behave around them and may inadvertently provoke an attack [30]. There were 54 cases of walkers being attacked by cattle in Great Britain between January 1993 and May 2013 [30], demonstrating that cattle can be a risk to anyone who may come in contact with them.

It has been reported that farmers could interpret their cattle’s facial expressions and posture to establish when an animal may become an immediate risk [35]. However, this may not be sufficient to reliably estimate the risk of an attack, as it has been reported that bull attacks can occur without any visual behaviour to communicate a warning [31]. This is supported by a study of 15 bull attacks, in which all victims reported that their attacks were unprovoked and that animals become aggressive suddenly [32]. Overconfidence on the part of the handler may also be a danger [31]. An example of this overconfidence may be in children who are overfamiliar with the farm (and animals/equipment therein) and may not perceive risks [33].

Attitude and behaviour can be improved through extension services and education [48]. The range of people who may come in contact with cattle includes individuals traversing farmland, living on a farm, and individuals working on farms. This list is not exhaustive and demonstrates the wide range of extensions required to reach individuals. Clear signage on farms and areas where cattle may be encountered could help alert visitors who are less familiar with cattle to the dangers. The extension should be targeted to suit the individual demographic, for example, farm safety days for children [49]. Multifaceted approaches for farm managers which include an environmental change and safety audit have been shown to be effective [50]. Another method targeted at farmers is a “fear appeal” which motivates a farmer to adopt safer behaviours by exposing them to a hypothetical threat situation and then providing information on how to mitigate the threat [33]. A survey to assess the role of fear appeals in motivating farmers to adopt safer practices found that for a fear appeal to work, the farmer must be able to identify a threat and carry out the mitigating action. Additionally, the farmer must believe that the danger is real and that it could happen to them [33]. It was reported that farmers are more likely to act on simple threats and that more complex threats which are harder to identify and/or had multiple causes may be ignored. It was reported that farmers thought that written rules were unnecessary and valued their own experience more than training [33]. Furthermore, farmers felt extension officers were inadequate and so may have ignored dangers highlighted by advisors, and instead of adopting the mitigation recommendations, the farmer-controlled his or her fear reaction by denying the existence of the threat [33].

#### 3.4.3. Facilities and the Environment

The design of facilities differs amongst farms [45], and there was geographical variation and a range of management systems employed in the studies included in this review. Dairy cattle require different facilities to carry out milking and they may be housed either continuously or for part of the year [51]. Many beef cattle outside Europe are raised on feedlots, however, the facilities in these lots may vary due to climatic conditions and rainfall [52]. In cool temperate areas, cattle may be housed during wet winters and graze outside during summer. Thus, farm design cannot be uniform and must be tailored to the management system employed, geographical topography, or regional variation. Poorly designed or inappropriate facilities are known to be significant contributors to cattle-related incidents [31]. The facilities dictate how the human interacts with the animal, how much time is spent in the risk zone, and thus how much scope there is for an incident to happen [44]. However, it is poorly understood which specific aspects of the facilities are most important. In-depth on-site investigations of a representative sample of cases of attacks have been suggested to be necessary to clarify the contribution of different aspects [31]. Minimizing personal contact time may be key, especially when working with bulls, as most bull-related incidents occurred when the person entered the bull’s territory [31]. Using suitable facilities rather than personal contact to perform part of the handling may aid in this. For instance, automatic manure scrapers can be used to “gently get cows to stand and move” [46], a task that would otherwise require a direct approach by the handler. Different facilities may be required for different animal types. For example, due to their size and strength, bulls are often held in a bullpen, and it is recommended to only enter a bullpen when the bull is restrained [31]. However, to improve the human–animal relationship it is recommended that interactions with dairy cows take place while the animal is unrestrained [40].

Apart from facilities directly designed to handle the animals, the comprehensive design of the farm and its management will also contribute to the risk of injury. Pastured dairy cows have been found to show reduced reactivity compared with cows in tie-stalls [41], potentially reducing the risk of injury. Conversely, beef animals that are grazed in an extensive system with little human contact are known to be more aggressive and evasive than cattle in more intensive systems [35,45].

In addition to the physical arrangement of the facilities, other external factors, such as the presence of a dog, can impact the occurrence of injury. Cattle may perceive dogs as more threatening than humans [30]. The presence of a dog was identified as a risk factor and recorded in 64.8% (n = 35) of attacks on walkers traversing farmland [30]. The effect of how the human interacted with the dog was less clear, with attacks occurring in situations both where the human had picked up the dog or released it from its lead. Other studies mentioned that the presence of a dog was recorded, but did not include this in the subsequent analysis [43].

#### 3.4.4. The Individual Animal

The animal breed was not often reported, however, bull attacks were predominantly inflicted by animals classified as dairy breeds [31]. Eight studies focussed on dairy cows, however, it is possible that this could be attributed to more intensive management on dairy farms compared with beef farms, offering more opportunities to observe handling. A focus group discussion found that farmers who bred Pyrenean cattle could not only distinguish the temperament of Pyrenean cattle from other breeds but could also identify variation in temperament between individual animals of the Pyrenean breed. Farmers who took part in the study attributed these differences within the breed to genetics, age, and sex [35]. A review of media sources and published literature reported that bulls caused more injuries than cows [30], and one study focussed exclusively on bulls because these are considered to be more likely to attack [31]. However, in addition to bulls, primiparous and freshly calved cows are perceived as very dangerous to work with by the staff of American dairies [37]. Whilst one author suggested that injuries caused by cows were more likely to be due to ‘nonintentional’ behaviour than targeted attacks [31], cows with calves are likely to exhibit maternal aggression when they perceive a threat to their calf [30]. Although maternal aggression was identified as a risk factor, it is seen as a desirable trait in extensive management systems where a cow may have to defend its young from predators [35].

Although there were no clear associations between sex and aggression, age was reported to affect animal aggression, with fewer attacks recorded from younger bulls [31]. Conversely, cows seem to become more docile with increasing age [35,37]. Despite this perception, maternal defensiveness of breeding beef cows when handled was reported to be a repeatable trait and remained consistent throughout different parities [38]. In addition to the inherent traits of the animal, the animal’s emotional state or how it has been previously treated can also affect the likelihood of an incident. The deviation from baseline heart rate was significantly greater when moving cows to hoof trimming than to milking, implying that it is more stressful for the animal [46], and more risk situations were observed during this apparently more stressful procedure [44]. Together, this suggests that stressed cows are potentially more dangerous. Hoof trimming is a less regular occurrence which cows will be less accustomed to than milking. Training cows to cope with aversive procedures can reduce fear levels. This can reduce stress for the animal [46], thus reducing the likelihood of behaviours that may result in injury. For example, heifers that had been introduced to the milking parlour 10 days prior to calving had a tendency to kick less often than ones that had not been introduced during their first post-partum milking [39]. Management or experiences during early life may impact an animal’s adult temperament. Certain actions in young bulls, such as rough play or teasing, can encourage aggression [31]. Similarly, offspring can inherit behavioural traits from cows through post-partum experiences and genetics [35].

The identification and slaughter of aggressive bulls have been recommended [31]. This would not only remove the threat from the dangerous bull but also prevent further dissemination of any genetic predisposition to attack humans. However, survey data shows that at least some farmers continue to breed with potentially dangerous bulls [33]. None of the papers examined genetic effects, however, farmers who claimed to include behaviour in their selection criteria had cattle with lower avoidance distances [43], indicating cattle were less reactive towards humans. This took the form of a semi-structured interview and did not state the specific behaviour selected for [43].

#### 3.4.5. Under-Reporting and Poor Records

Despite the aforementioned importance of handling facilities and environment, the location of incidents is often reported as “Farm”, making it difficult to identify the specific facilities or location involved in a bull attack [31]. Additionally, the accuracy of any existing records depends on incidents being reported. Interviews with immigrant workers on US dairy farms found they were often afraid to report incidents as they may lose their jobs and risk deportation [37]. This, coupled with the one-sided reporting where the interaction is not observed but reported retrospectively, may lead to sensationalism. This is exemplified by media reports portraying the human involved as blameless and cattle depicted as unprovoked instigators of the attacks [30]. Many incidents are believed to be unreported if not fatal [30,31]. In a review of 287 incidences, it was found that 57% of the reported attacks resulted in a fatality, the rest had varying degrees of injury, and few had no injury reported [31]. To calculate the risk of an attack when encountering cattle, both the number of attacks and the total number of interactions are required, but full records for both are usually unavailable [30], with similar poor records of on-farm encounters [29]. In the absence of this information, the creation of a central database to encourage self-reporting of cattle attacks is recommended [30]. It is important to note the under-reporting may have led to publication bias, whereby interactions that led to smaller injuries are not included in the reviews. Furthermore, there is disparity between an injurious event and the reporting, with injury and fatality data often not providing information on why livestock-handling injuries occur [53]. Linking injury to an event on-farm is made more difficult as there is no internationally accepted reporting method [2].

The review papers are only representative of the records they reviewed, and both stated that records available for review may not have reported all incidents [30,31]. Media will be more likely to publish incidents where an injury has occurred, with ‘near misses’ not reported on. There is also the likelihood that incidents reported retrospectively by the human may not be correct as the human involved will not want to admit they behaved irresponsibly. The main focus of a review of hospital admissions was the outcome of the attack [32]. In all cases, patients reported that the bull attacks were unprovoked, but there is no evidence to support or oppose this. It is possible the injured party did not accurately describe the interaction leading to injury, or they may have been unable to interpret the cattle’s behaviour as threatening.

The papers in this review were limited as they depended upon retrospective reports, farmer opinions, or small studies. To truly understand all factors which lead to cattle-induced injuries, it would be necessary to undertake multiple surveillance studies which include the observation of multiple human–animal interactions under various management systems and using different facilities. To undertake this surveillance at the necessary scale to record all potential human–animal interactions under all potential environmental conditions, the costs and manpower required would prove prohibitive.

#### 3.4.6. Mitigation of Dangerous Interactions

There is a range of recommendations to mitigate bull attacks during the handling of cattle: ensure the bull wears a ring in its nose, have appropriate facilities, do not handle a bull older than one year old alone, ensure beef bulls are not crowded, and make sure to be wary of where a bull is at pasture [31]. These recommendations were taken from a single extension article published in 1981 by the College of Agricultural and Life Sciences, University of Wisconsin. Twenty-four separate web pages, which had guidelines for safe behaviour when encountering cattle when walking through fields of cattle, were reviewed [30]. These were collated and gave some contradictory advice such as “keep dog under control” and “let dog off the lead”; the National Farmers’ Union of Wales advised “Be bold and walk straight through the cattle if they come towards you”, whereas several other sources recommended skirting around the edge of the herd. In addition to the contradictory advice, some of the advice is less than practical, such as twisting the ring in a bull’s nose, as this would require being in very close proximity to the animal and not taking evasive action.

Farmers were reported to feel that the presence of horns makes an animal more dangerous even if the individual animal is not aggressive [35]. Dehorning (‘polling’) cattle can reduce the damage resulting from the interaction. However, herds with a higher proportion of cows that had undergone polling were more fearful of humans with a higher avoidance distance and a higher proportion of fearful cows during tactile interaction [42]. A survey undertaken found that the main reason for dehorning was reported as fear of injuries to other animals (95%), farmers (84%) family members (48%), visitors (8%), or employees (5%) [34]. Twelve farmers reported injuries caused by cows with horns (five of which were injuries inflicted on humans, four of which were inflicted on other animals, three of which were encounters with dead animals).

Cows are generally docile but are known to be dangerous when agitated [46]. Despite this, focus groups discussions found that only 50% of farmers agreed with the statement cattle are dangerous [43]. Risk management procedures recommended for the protection of third parties often concentrate on controlling cattle, with likely detrimental effects on the farmer. Suggestions to reduce injury to walkers included removing herds from publicly accessed fields, or euthanasia of reactive cattle with subsequent cost and management implications for farmers [30]. The review of cattle attacks on people traversing farmland recommends that bulls that attack should be killed to prevent further attacks [30]; this would incur a cost for the producer. When hazards were reported by migrant workers, the manager often did not act [37]. Health and safety regulations are in place for working on the farm, however, some farmers fear the introduction of additional regulations as they may lead to unnecessary costs [33]. Farmers were reported to be more afraid of risks to others, but more likely to take action on threats that were personal (i.e., affected the farmer themselves), and to blame others for being irresponsible [33]. Although the measures are taken to mitigate risks to farmers, their staff and third parties on the farm may be seen as costly to the farmer. It is necessary to manage cattle to prevent the risk of injury for the handler’s safety, but also the potential compensation costs if staff are injured [37].

### 3.5. Limitations of This Review

This study concentrated on human interactions and management which led to behaviours in cattle that could potentially cause injury. The search was targeted at papers published between 2000 and 2021 and concentrated on *Bos taurus* animals. Although broadening the search would have increased the number of papers for review, it would not have provided contemporary information for the farm systems and cattle relevant to the production systems in the legislative area targeted by the policy the paper aims to inform. Four papers were discarded as they were not published in English or the full text could not be accessed, however, the abstracts were available. Of the two non-English texts, one was a German study that carried out a questionnaire with 317 farmers, of whom 25% reported injuries caused by bulls [54], however, the information could not be assimilated into this review in the absence of the full text. The other paper discarded due to language may not have been relevant as it surveyed workers on the practices of dairy farms in Brazil [55], but the contents of the survey could not be assessed as to whether it would involve animal interactions or not. The two texts which could not be found were not full articles but conference texts. This review was not limited to full papers and included conference texts as it can reduce publication bias [56], but they may be less reliable as they are not subject to peer review [56]. The inclusion of the discarded papers may have added value to this review, however, it is likely that they may have added to the discussion within one of the six themes rather than introducing a new theme. The six themes identified in this review were interlinked. It is not possible to quantify the importance of each theme, thus none of these themes can be considered in isolation and must be accommodated in the management system employed when working with cattle.

## 4. Conclusions

A systematic review was developed searching for studies that identified how human actions and management systems could impact animal behaviour and increase the risk of injury to a person. Seventeen papers were identified which met the criteria. Six main themes were identified: actions of humans; human demographics, attitude, and experience; facilities and the environment; animal involved; under-reporting and poor records; and mitigation. The under-reporting of incidents, variation in management methods, and contradictory mitigation advice make it difficult to develop international standards of best practice. To develop appropriate policy to reduce injuries caused when working with livestock, it is recommended that a better understanding of causes of injury is attained either through observational studies or surveys which identify risk factors. To reduce cattle behaviours that can potentially induce injuries, it is recommended that standardised reporting and recording of incidents be introduced. Collation of injury data and the human actions resulting in injury will clearly identify the behaviours and facilities which increase injuries and so can inform choices to mitigate further incidents. Guidance for those working with cattle must be practicable, easily accessed, and there must be a consensus amongst providers to ensure that contradictory guidelines are avoided. This could lead to a specialized training programme for those working with cattle. Guidance must also be provided to anyone who may come into contact with cattle. This could be implemented through appropriate signage for those traversing farmland or visiting farms or extension through schools to target children living on the farm. It is further recommended that facilities that improve farm safety are installed and adequately maintained. These recommendations should be supported by including temperament and behaviour traits in breeding goals and the systematic culling of dangerous animals.

## Figures and Tables

**Figure 1 animals-12-00776-f001:**
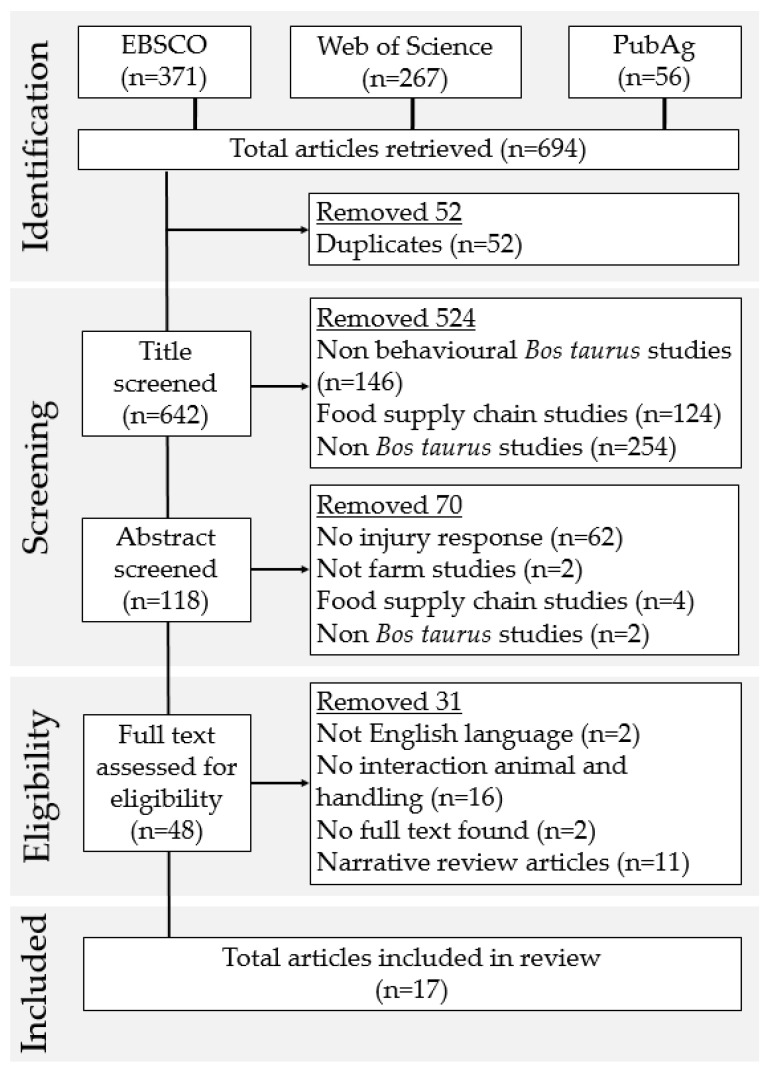
Article selection process detailing the number of articles included and excluded at each step of the review using the PRISMA (Preferred Reporting Items for Systematic Reviews and Meta-Analyses; [27]) guidelines.

## Data Availability

Not applicable.

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
