# Peer review of "Human–Animal Interactions with Bos taurus Cattle and Their Impacts on On-Farm Safety: A Systematic Review"

_animals, 2022, doi:10.3390/ani12060776_

Round 1

Reviewer 1 Report

General comments: 
In this manuscript the authors did a systematic review on the relationships between humans and Bos taurus cattle associated to the risk of causing injuries in humans, aiming to use it outcomes to develop guidelines and identify where improvements in animal handling need to be made. Despite of being well defined, it seems to me that the objectives were not achieved, since I did not find any clear guideline or description about how to improve animal handling to reduce the risk of labour related accident.

Additional comments and suggestions are presented below.

Additional comments

Title
Since the review is limited to Bos taurus cattle, this should be informed in the title.

Introduction:
L70-72. Why to limit to the papers published from the year 2000, if some good recommendations about safe cattle handling procedures were developed before this year? Justify.
L85. Where is the “target legislative area”? Inform.

Material and methods
Figure 1. I suggest removing Figure 1, since all information shown in it are repeated in L149-158 or, alternatively, consider removing the sub-section 3.1 paragraph (L149 -158).

Results and discussion
L480-488. I did not understand why to include these information here, since they cannot be characterized as manuscript limitations. Think about.

Conclusions
L497. Replace "to identify" to "searching for".

Reviewer 2 Report

1. Line 124, say "be written in English". The reason why only works written im english have been analised is not explained.

 2. In the figure 1, paragraph "Sreening", it says 7 and should say 70.

Reviewer 3 Report

The article A systematic review of human-cattle interactions and their impacts on on-farm safety, meets the formal requirements for printing. There are no self-citations and the manuscript is carefully prepared.

 At first glance, the study seems interesting for farm workers, breeders, veterinarians and scientists. However, when reading it, it turns out that out of an impressive number of scientific reports, 17 were eventually selected for review.

Meanwhile, in the opinion of the reviewer, the report has several errors:

  1. In the opinion of the Reviewer, drawing conclusions on a relatively small group of reports is unjustified. In fact, the conclusions add nothing new to science.
  2. The authors do not support the data with any officially recorded data. It seems that these 17 literature reports could be enriched or compared with the official data collected by state institutions dealing with the collection of statistical data.
  3. There is no differentiation of cases depending on the country (in different countries, the standards and types of breeding, and thus animal behavior and handling may be different).
  4. In such a small number of cases selected for the study, interactions with other large animals could be included.

Reviewer 4 Report

Authors carry out a systematic review of literature relating to farm management practices to assess the factors which may lead to a dangerous interaction with cattle.

The title indicates the aim of the manuscript and the abstract is well written. The introduction is also well written. The objectives of the study are of interest and are in line with the scope of the journal.

The manuscript is well organized. The abstract clearly indicates the work objective, methodology and result of the study. The methodology is well articulated and the description is well made.

The conclusions are consistent with the evidence and arguments presented.

The reference is appropriate.

In my opinion, the manuscript could be accepted for publication.

Author Response

Dear Reviewer 4,

I appreciate you taking the time to read my manuscript and for your valued response.

Many thanks

Round 2

Reviewer 3 Report

Thank you for answering my doubts, I am satisfied. I accept the amendments and have no more doubts.